# Subpicosecond metamagnetic phase transition in FeRh driven by non-equilibrium electron dynamics

Federico Pressacco [1,2✉], Davide Sangalli[3,4], Vojtěch Uhlíř [5,6], Dmytro Kutnyakhov [2], Jon Ander Arregi [5], Steinn Ymir Agustsson [7], Günter Brenner[2], Harald Redlin[2], Michael Heber[2], Dmitry Vasilyev[7], Jure Demsar [7], Gerd Schönhense [7], Matteo Gatti[4,8,9], Andrea Marini [3,4], Wilfried Wurth[1,2] & Fausto Sirotti [9,10]

Femtosecond light-induced phase transitions between different macroscopic orders provide the possibility to tune the functional properties of condensed matter on ultrafast timescales. In first-order phase transitions, transient non-equilibrium phases and inherent phase coexistence often preclude non-ambiguous detection of transition precursors and their temporal onset. Here, we present a study combining time-resolved photoelectron spectroscopy and ab-initio electron dynamics calculations elucidating the transient subpicosecond processes governing the photoinduced generation of ferromagnetic order in antiferromagnetic FeRh. The transient photoemission spectra are accounted for by assuming that not only the occupation of electronic states is modified during the photoexcitation process. Instead, the photo-generated non-thermal distribution of electrons modifies the electronic band structure. The ferromagnetic phase of FeRh, characterized by a minority band near the Fermi energy, is established 350 ± 30 fs after the laser excitation. Ab-initio calculations indicate that the phase transition is initiated by a photoinduced Rh-to-Fe charge transfer.

[1] The Hamburg Centre for Ultrafast Imaging, Hamburg University, Hamburg, Germany. [2] Deutsches Elektronen-Synchrotron DESY, Hamburg, Germany. [3] Istituto di Struttura della Materia—Consiglio Nazionale delle Ricerche (CNR-ISM), Division of Ultrafast Processes in Materials (FLASHit), Monterotondo Stazione, Italy. [4] European Theoretical Spectroscopy Facility (ETSF) https://www.etsf.eu/. [5] CEITEC BUT, Brno University of Technology, Brno, Czech Republic. [6] Institute of Physical Engineering, Brno University of Technology, Brno, Czech Republic. [7] Johannes Gutenberg-Universität, Institute of Physics, Mainz, Germany. [8] LSI, CNRS, CEA/DRF/IRAMIS, École Polytechnique, Institut Polytechnique de Paris, Palaiseau, France. [9] Synchrotron SOLEIL, L'Orme des Merisiers, Gif-sur-Yvette, France. [10] Physique de la Matiére Condensée, CNRS and École Polytechnique, IP Paris, Palaiseau, France. ✉email: federico.pressacco@desy.de

Emergence of long-range ordered states in condensed matter is typically a consequence of a fine interplay between the coupled spin, charge, orbital, and lattice degrees of freedom[1–4]. The mechanisms vary between different correlated oxides and metallic systems leading to specific dynamical behavior. Excitation with ultrashort electromagnetic pulses offers the most efficient means to control the physical properties of condensed matter systems on a femtosecond time scale[5–7]. Materials featuring first-order phase transitions (FOPTs) with abrupt changes in their order parameters are especially appealing for ultrafast devices based on a functionality switch. In this regard, prominent examples are the insulator-metal transition in $VO_2$[8,9] or $1T–TaS_2$[10,11].

In order to obtain a good understanding of the relevant mechanisms triggering the transition, it is necessary to explore the fundamental timescale of FOPTs. However, this is often challenging with multiple coupled degrees of freedom displaying complex dynamics upon laser-induced excitation. Moreover, macroscopic phase coexistence at the FOPT, specifically, the processes of nucleation and domain growth, complicate the disentanglement of dynamic changes in order parameters. The fundamental question, whether the modification of electronic structure drives the transition, is extensively debated in the literature, giving key arguments on the role of photoexcited states in double-exchange interactions[12], electronic precursors closing the insulating gap[13], or the existence of intermediate transient phases[2,9,14,15].

In the case of ferromagnetic materials, magnetization dynamics triggered by a laser pulse leads to ultrafast demagnetization[16], associated with changes in the spin polarization[17–20], which is impacted by the spin-dependent mobility of electrons[21–23]. In contrast, disentangling the ultrafast response of coupled order parameters of magnetic FOPTs has been far less investigated. Ultrafast generation of ferromagnetic (FM) order has been observed so far in a relatively small group of materials such as manganites[1,24] or $CuB_2O_4$[25]. Understanding the phenomenon of laser-induced subpicosecond generation of FM order across a FOPT is still a major challenge in femtomagnetism[26].

In this work, we focus on FeRh, a metallic material that undergoes a metamagnetic FOPT from antiferromagnetic (AFM) to FM order at $T_M \sim 360$ K and exhibits coupled structural, magnetic and electronic order parameters[27,28] (see Fig. 1a). The thermally induced, quasi-static phase transition in FeRh (depicted in Fig. 1b by the green arrow) has been extensively studied by following the sample magnetization, lattice parameter or resistivity[29–32]. Moreover, numerous works have studied the AFM-FM phase transition by means of time-resolved techniques, where photoexcitation above a threshold intensity results in a nonzero net magnetization. Seminal pump-probe magneto-optical studies of FeRh films suggested subpicosecond generation of FM order[33,34]. Bergman et al. later suggested a scenario in which FM domain nucleation occurs at ultrafast time scales, but the subsequent establishment of long range magnetic order is naturally slower, a process consisting of FM domain growth, coalescence, and magnetic moment alignment[35]. Subsequent research efforts tracking the net magnetic moment correspondingly found a much slower transition on the order of several picoseconds[35–37]. Thus, detection techniques susceptible to the magnetization direction result in a perceived delay in the emergence of FM order. Besides, time-resolved x-ray diffraction experiments indicated that the speed of the transition might be set by the time scale of the structural changes and therefore limited by the speed of sound (~5 nm/ps), so that magnetic and structural order emerge concurrently[37,38].

However, FM order can also be traced by directly exploiting the specifics of the electronic structure, naturally manifested in terms of spin unbalance and the appearance of majority and minority spin bands[39–41]. This is independent of spin alignment along a particular direction, and thus allows inspecting FM order via x-ray photoelectron spectroscopy (XPS)[42–44]. Similar to the electronic signature of the insulator-metal transition[8,10], it was demonstrated that the modification of electronic bands might prove equally useful to investigate the laser-induced generation of FM order across the magnetic FOPT in FeRh[45].

Here, utilizing time-resolved photoelectron momentum microscopy and supported by first principle calculations, we demonstrate that it is the light-induced modification of the

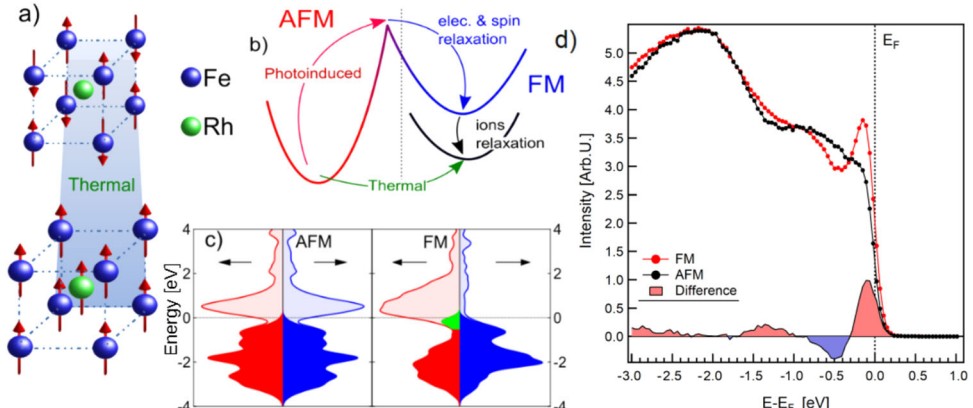

**Fig. 1 Electronic properties of FeRh across the metamagnetic phase transition. a** Sketch of the isostructural metamagnetic phase transition in FeRh. At room temperature (top) the system is AFM showing atomic magnetic moments only at the Fe atom sites ($m_{Fe} = \pm3.3\mu_B$). Above 360 K (bottom) the system has ferromagnetically coupled magnetic moments at the Fe ($m_{Fe} = 3.1 \mu_B$) and Rh ($m_{Rh} = 0.9 \mu_B$) sites. The whole unit cell expands isotropically by about 1% in volume[69]. **b** Schematic representation of the two possible paths in the AFM to FM phase transition: direct thermally-driven transition to the FM phase (green arrow), and two-step transition going through a transient electronic state reached during photoexcitation (red arrow) followed by relaxation to the equilibrium FM state (blue and black arrows). **c** Calculated spin-resolved electronic density of states in the AFM and FM phases. The filled areas represent the electronic occupation at thermal equilibrium, with the green area highlighting the position of the Fe minority band (see manuscript text). **d** Measured x-ray photoelectron spectra of FeRh in the AFM (black dots) and FM (red dots) phase. The solid curve at the bottom of the graph is the difference between the two spectra, which allows to appreciate the relative electron density change across the transition. The data correspond to quasi-static thermal cycling experiments prior to the time-resolved measurements.

electronic band structure that triggers the phase transition in FeRh. In particular, we show that ultrafast laser excitation induces a charge transfer between the Rh and Fe atoms, serving as a non-equilibrium precursor for the formation of the FM band structure on the subpicosecond timescale.

## Results

The sample under study consists of an epitaxial 80-nm-thick FeRh(001) film grown on a single-crystal MgO(001) substrate, undergoing the AFM-to-FM phase transition at 377 K (see Supplementary Fig. 1). The establishment of the FM phase in FeRh is accompanied by the appearance of a narrow peak in the electronic density of states, located about 150 meV below the Fermi energy $E_F$, as a result of the occupation of a spin-polarized Fe band (see green highlighted area in Fig. 1c). The photoelectron spectroscopy data shown in Fig. 1d are the momentum integrated energy distribution curves measured at room temperature for the AFM phase (black dots) and at 420 K for the FM phase (red dots). We use this spectral feature to follow the emergence of the FM phase after laser excitation. Pump-probe experiments were performed at the FLASH Free Electron Laser (FEL) in Hamburg using near-infrared (800 nm, 1.55 eV) pulses of 90 fs coupled with 130 fs soft x-ray pulses with a photon energy of $\hbar\omega = 123.5$ eV (see "Methods" for details).

Figure 2a presents the time-resolved, $k$-integrated photoelectron spectra measured as a function of the delay between the optical pump and x-ray probe pulses, focusing on a 4 ps window around time zero $t_0$. The laser fluence in this pump-probe experiment was 5.6 mJ/cm², which is above the threshold value to induce the FOPT (see Supplementary Fig. 2 for additional experiments at different laser fluences). One can clearly identify distinct regions in the time-dependent spectra: the temporal overlap between the optical and x-ray pulses (about 100 fs around $t_0$), the relaxation of electrons toward the Fermi energy on the 100 fs timescale, and the subsequent changes in the density of states near the Fermi level, associated with the formation of the Fe minority band. Differential energy-dependent profiles, reported in Fig. 2b, provide a clearer picture. These are retrieved by averaging the measured photoelectron spectra within a ±50 fs temporal region for each of the indicated time delays, and subtracting the average photoelectron spectra at negative time delays.

The photoexcited electrons relax via electron-electron and electron-phonon scattering, leading to the onset of a Fermi-like distribution (see the red shaded areas for $E - E_F > 0$ in Fig. 2b). A reduction of the electron density below the Fermi level is also observed around $t_0$ (blue shaded areas for $E - E_F < 0$ in Fig. 2b). At the same delay, the spectrum shows excitation of electrons up to 3.1 eV above $E_F$ (see also the inset in Fig. 3). Note that the electron density above 1.5 eV is almost two orders of magnitude lower. The changes in the population above the Fermi level can be explained by one- and two-photon absorption processes, such that laser excitation ($\hbar\omega_p = 1.55$ eV) promotes electrons into states within the energy range $[E_F, E_F + 2\hbar\omega_p]$. Then, electrons start to relax toward the Fermi level and accumulate in an energy region of a few hundred meV above $E_F$. On the other hand, the transient depletion of electronic density below the Fermi level is concentrated between −3.1 eV and −1.55 eV $[E_F - 2\hbar\omega_p, E_F - \hbar\omega_p]$. Assuming the photo-induced depletion, one would expect to observe measurable changes in the photoelectron yield only for energies 1.55 eV below the $E_F$ (as the two-photon contribution should be negligible). This suggests that the laser induced changes in the electronic distribution close to the Fermi level cause severe modifications of the deeper lying bands, meaning that the in-fieri FOPT involves changes in the overall electronic structure of the system.

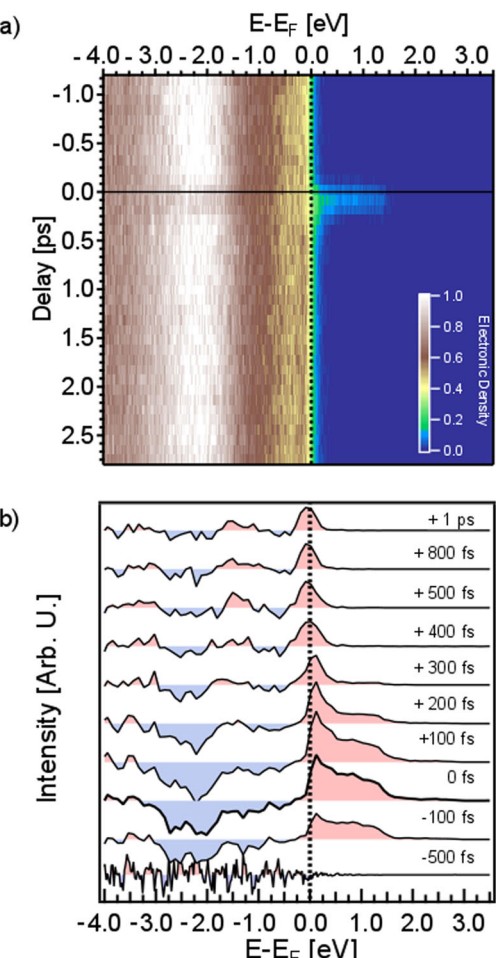

**Fig. 2 Time-resolved x-ray photoelectron spectroscopy of FeRh. a** Energy- and delay-dependent matrix of the measured spectra. The vertical dashed line marks the position of the Fermi level $E_F$, while the horizontal solid line designates the time zero, $t_0$. The appearance of electronic density in the unoccupied states is a fingerprint of the laser excitation. We used this spectral feature to identify the temporal overlap between the optical pump and the x-ray probe pulses. **b** Differential photoelectron spectra at selected delays. To enhance the signal to noise ratio, we average the unpumped spectra (between −1 ps and −0.5 ps) and subtract the average spectrum from each row of the matrix. This allows evaluating the statistical noise at negative time delays (−500 fs) as well as accentuating the temporal evolution of the photoelectron spectra. Red and blue shaded areas indicate an increase and a reduction of the electron density with respect to the spectra at negative delays, respectively.

To elucidate the electronic dynamics near the zero time delay, we compare the experimental results with time-dependent density functional theory (TD-DFT) calculations of the electronic structure performed on FeRh (see Fig. 3). Here, we select a laser fluence which gives a good quantitative agreement with the measured photoelectron spectra (see "Methods"). The calculated one- and two-photon absorption intensity above the Fermi level presented in the inset of Fig. 3 corresponds to an excitation of 0.25 electrons per FM unit cell of FeRh (see Fig. 1a), i.e., ~$10^{22}$ cm$^{-3}$. This is close to the experimental estimate of 0.15 electrons per unit cell for a 5.6 mJ/cm² fluence. In Fig. 3, the calculated XPS spectrum and its time evolution are obtained considering (i) the time evolution of the electronic distribution function $f_{n\mathbf{k}}(t)$, and (ii) the time evolution of the electronic band structure $\epsilon_{n\mathbf{k}}(t)$ (see "Methods"). The changes in $f_{n\mathbf{k}}(t)$ can give a significant signal only

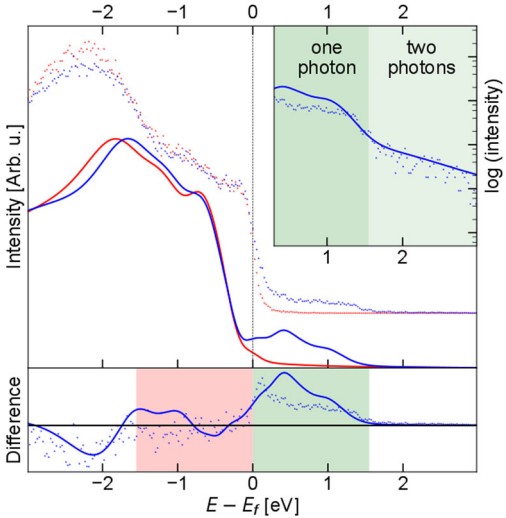

**Fig. 3 Comparison of experimental and computed photoelectron spectra of FeRh.** Experimental spectra (dots) at equilibrium ($t = -0.5$ ps, red) and at the maximum of the pump pulse ($t = 0$ ps, blue) are compared in linear (top panel) and logarithmic scales (inset) to the TD-DFT calculations (solid lines) of the electronic structure of FeRh in the AFM phase (red) and at laser excitation (blue). The bottom panel displays the difference between the corresponding experimental and theoretical spectra in the top panel. The green (red) region represents the energy range to (from) which electrons are excited with one-photon (1.55 eV) processes. The light green region in the top panel inset identifies the energy range that can be reached by two-photon (3.1 eV) processes.

in the red and green shaded regions between $-1.55$ eV and $+1.55$ eV, since the electron redistribution caused by two-photon absorption process is negligible. Thus, the signal below $-1.55$ eV is fully due to changes in $\epsilon_{n\mathbf{k}}(t)$. The fact that it is negative implies a reduction in the density of states (see the relative height difference of the red and blue solid curves in Fig. 3). On the other hand, the changes in the region between $-1.55$ eV and 0 eV result from two effects that sum up to a negligible signal. $f_{n\mathbf{k}}(t)$ gives a negative contribution due to the promotion of electrons to states above the Fermi level. For the overall signal to be negligible, the change in $\epsilon_{n\mathbf{k}}(t)$ must result in an increase of the density of states which compensates the reduction in the occupations. Such compensation is almost exact in the experimental data, leading to an overall signal that appears as a reduction of the broad peak at $\approx -2$ eV. The same compensation is only partial in the simulation leading to an overall signal that appears as a shift of the same peak. The region above the Fermi level seems to be mainly governed by $f_{n\mathbf{k}}(t)$. The overall agreement between theory and experiment is very good. The observed slight differences may be due to relaxation processes in the experiment already active during the pumping phase, making the electron distribution more peaked toward the Fermi level.

The relaxation of excited electrons proceeds with the formation of a peak above the Fermi level between 100 fs and 300 fs (see Fig. 2b). At a time delay of around 400 fs, the peak crosses the Fermi level and after 500 fs, its position stabilizes at the energy value which is characteristic for the Fe minority band of the FM phase in FeRh[45]. We also observe a modest but perceptible decrease (increase) of the spectral weight around $-0.5$ eV ($-1.4$ eV) binding energy for $t > 0.3$ ps, a characteristic that matches the difference of the static AFM and FM photoelectron spectra in Fig. 1b. Additional pump-probe experiments at a slightly lower laser fluence of 4.5 mJ/cm$^2$ show entirely equivalent electron

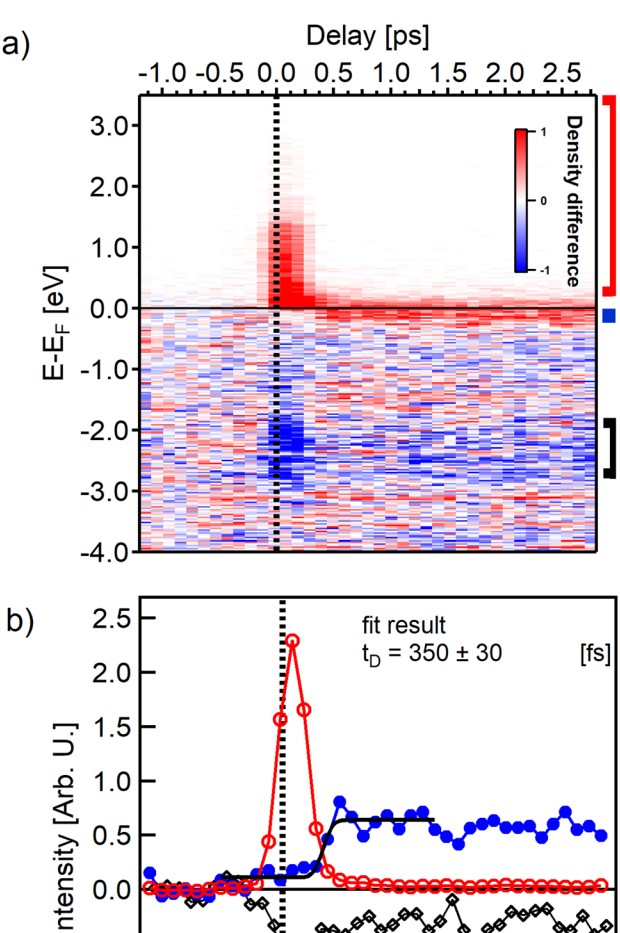

**Fig. 4 Subpicosecond generation of the electronic FM order in FeRh. a** Energy- and delay-dependent differential matrix of the measured photoelectron spectra. We subtracted the average of unpumped spectra (between $-1$ ps and $-0.5$ ps) from each measured spectrum. The effect of laser excitation is evident around time zero, indicating depletion of the occupied states and the corresponding population of the empty states. In addition, an increase in the electronic density close to the Fermi level is observed at positive time delays. The red, blue, and black bars on the right hand side mark the representative integration regions for tracking the electronic dynamics in the unoccupied states, the formation of the Fe minority peak, and the modification of the deeper bands, respectively. **b** Temporal evolution of the electronic density in the three characteristic energy regions marked in (**a**). The population of states above the Fermi level shows a fast rise and a consecutive decay around $t_0$ (empty red circles). The deeper bands (empty black diamonds) show a corresponding depletion and recovery which reflects the dynamics of the unoccupied states. However, their occupation level stabilizes 300 fs after $t_0$ and remains constant thereafter. The electronic density slightly below $E_F$ (filled blue circles) shows a moderate increase during the laser excitation up to 300 fs delay, followed by a pronounced increase due to the shift of Fe minority band below $E_F$ at a delay $t_D$ of 350 fs. This value was extracted by fitting the error function to the experimental results (black solid line). Subsequently, the minority Fe band peak intensity remains constant throughout the investigated delay range.

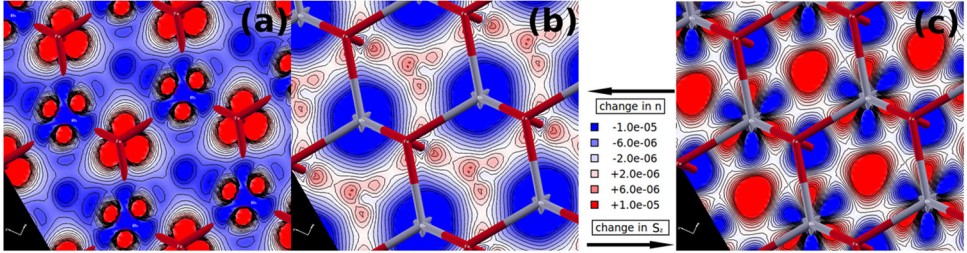

**Fig. 5 Computed photoinduced changes in charge and spin density in FeRh. a, b** Changes in the charge density $n(\mathbf{r}, t_f) - n^{eq}(\mathbf{r})$ and (**c**) spin density $S_z(\mathbf{r}, t_f) - S_z^{eq}(\mathbf{r})$ at the end of the laser pulse ($t = t_f$) with respect to the equilibrium configuration. Densities are in atomic units ($[e/a_0^3]$ and $[\mu_B/a_0^3]$, where $a_0 = 5.29 \times 10^{-11}$ m is the Bohr radius. 0.25 electron per unit cell corresponds to a density of $7 \times 10^{-4}$ $e/a_0^3$.) on two different FeRh{111}-planes: the plane just below the $Fe(\uparrow)$ atoms (**a**) and the plane containing the Rh atoms (**b, c**). Fe and Rh atom positions are indicated by red and gray segments of the lattice, respectively. Charge is transferred from the occupied (bonding) orbitals of Rh atoms (panel **b**), to the unoccupied (anti-bonding) orbitals of Rh and Fe (**a**). Since the Fe anti-bonding levels are filled in this process, the local Fe spin density is reduced. As a result, there is a redistribution of spin density around the Rh atoms and a reduction of the local spin moment at the Fe atoms (**c**). The movies in the Supplementary Information show the simulated time evolution of the charge density variations at the Fe and Rh sites as represented in (**a**) and (**b**) at the beginning and the end of the laser pulse.

dynamics in terms of the pump-induced increase in the density of states above $E_F$, the subsequent relaxation of electrons and the subpicosecond formation of the minority band (see Supplementary Fig. 2). The fingerprint of the FOPT for laser fluences above ~3 mJ/cm² remains 20 ps after the laser pulse (see Supplementary Fig. 2) and beyond (see ref. [45]). Photoexcitation at fluences below this threshold gives rise to transient changes in the electronic occupation landscape, but does not cause a persistent modification of the band structure (see Supplementary Fig. 2).

Further insights into the FOPT dynamics are obtained from the complete differential matrix, presented in Fig. 4a, monitoring the modification of the electronic density at different time delays. We selected three energy ranges marked by the red, blue, and black bars placed on the right hand side of Fig. 4a. The first (red) accounts for the electronic density above 200 meV and identifies the total number of electrons injected into the unoccupied states upon photoexcitation. The second (blue) goes from −240 meV to 0 meV and is used to monitor the formation of the Fe minority peak across the phase transition. The third (black) includes the region from −2.8 eV to −1.8 eV and is used to follow the modification of deeper bands.

Figure 4b shows the time evolution of the integrated signal in each range. The injection of electrons into the unoccupied states takes place near zero delay (we recall that our system's response function is 150 fs) and then rapidly decays (red empty circles), the process being finished by about 500 fs. The intensity at the position of the Fe minority peak starts to grow already during the laser excitation (filled blue circles), due to both the modification of the electronic structure and the Fermi level smearing. More specifically, this modest intensity growth starting at $t = 0$ is linked to the presence of an instantaneous increase in the density of states peaking right above the Fermi level (see Fig. 2b, $t = 0$ ps) and can be considered as a precursor of the metamagnetic phase transition. However, it is only about 300 fs after the excitation that the electronic density corresponding to the minority band peak displays a sharp increase, associated with a persistent band structure modification. The characteristic time delay $t_D$ and subsequent rise time $\tau$ of the transition are obtained by fitting the integrated electron density at the Fe minority peak region with an error function (see "Methods") and yields values of $t_D = 350 \pm 30$ fs and $\tau = 220 \pm 110$ fs. In addition, the data represented by black diamonds in Fig. 4b, which indicate the time-dependent population of bands in a region below $E_F$, nearly mirror the population of the unoccupied levels up to 300 fs, with a fast depletion and recovery. The curve stabilizes thereafter at a finite value, implying permanent modifications of the deeper bands already after 300 fs, during which the Fe minority peak is still shifting toward its final

position at 150 meV below $E_F$. Similar differences in the behavior of the electronic density at and below the Fermi level have been observed during ultrafast demagnetization process in Fe[17] and Co[18]. In the present case, this behavior shows that the FOPT in FeRh is mediated by a transient electronic phase in which the electronic structure is different from both the AFM and FM phases. This precursory phase exists in a delay range from time zero up to 500 fs after laser excitation.

## Discussion

The unexpected reduction of the photoelectron yield below the one-photon absorption range (~1.55 eV), an energy region where the electronic populations $f_{n\mathbf{k}}(t)$ cannot be strongly affected by the laser excitation, is explained by theoretical calculations. The effect can only be accounted for by considering the photoinduced change in the electronic band structure (i.e., density of states). Time-resolved XPS spectra are usually described and interpreted in terms of changes in the population $f_{n\mathbf{k}}(t)$ only. This is clearly insufficient for the ultrafast dynamics in FeRh, since changes in $\epsilon_{n\mathbf{k}}(t)$ must be considered on the same level even before the phase transition is complete.

The changes in $\epsilon_{n\mathbf{k}}(t)$ during and after photoexcitation provide insights into why FeRh would relax toward the FM phase. The photoexcitation process in the AFM state depletes the valence states, which are characterised by hybridized Fe-Rh bonding states, and fills the unoccupied states, where empty Fe "local minority-spin" anti-bonding states are mainly available. As a result two processes occur: a charge (and spin) transfer from Rh to Fe ($Rh \rightarrow Fe$), and a symmetric spin transfer from Fe "local majority" $\rightarrow$ Fe "local minority" ($Fe(\uparrow) \leftrightarrow Fe(\downarrow)$). Simulated charge and spin density changes upon photoexcitation clearly show this, Fig. 5. Here, the blue regions around the Rh atoms indicate a charge depletion, which can only partially be explained by a local redistribution. Most of the charge is transferred to the Fe atoms (see Fig. 5a), while there is a strong charge depletion around Rh atoms (see Fig. 5b). The $Rh \rightarrow Fe$ process also increases the local spin density, along the $Rh - Fe$ bonds, around the Rh atoms (see Fig. 5c). On the other hand, the $Fe(\uparrow) \leftrightarrow Fe(\downarrow)$ is a pure spin transfer process, with a strong reduction (~10%) of the local momentum of the Fe atoms, which is transferred into the vicinity of the Rh atoms (see Fig. 5c).

In the AFM phase, the zero magnetic moment around the Rh atoms is a result of a non-vanishing spin density that integrates to zero because of the hybridization with surrounding Fe atoms with opposite moments[3,41,46]. The photoexcitation alters this delicate balance. As shown for the magnetic disorder associated with the temperature increase, the decrease of the Fe-Fe first-neighbor

AFM couplings favors the FM order of the Fe subsystem, while inducing magnetic fluctuations on the Rh sites[3]. In turn, the induced Rh magnetic moments stabilize the FM over the AFM state[3,33,47,48]. Our simulations therefore suggest that the change of Fe-Rh hybridization ($Rh \rightarrow Fe$ process) and the optically induced intersite spin transfer (OISTR, $Fe(\uparrow) \leftrightarrow Fe(\downarrow)$ process) play a crucial role in the photo-induced transition. The OISTR process has been recently proposed, on the basis of TD-DFT simulations, as a key mechanism also in other multicomponent magnetic materials[49–51]. Here, it causes to weaken the AFM ordering but is not sufficient to trigger the magnetic transition alone (just after the photoexcitation, the system is still in the AFM phase).

The theoretical simulation, not including dissipating effects, cannot describe the dynamics after the photoexcitation when electron-electron, electron-phonon and electron-magnon interactions are at play, and the actual phase transition takes place. Taking into account dissipating effects, one would expect further dynamics of both $f_{n\mathbf{k}}(t)$ and $\epsilon_{n\mathbf{k}}(t)$: the formation of a Fermi distribution and its subsequent cooling (for $f_{n\mathbf{k}}(t)$), and the formation of the FM band structure (for $\epsilon_{n\mathbf{k}}(t)$). Instead, our experimental time resolution is fast enough to allow the identification of a bottleneck time in this process, i.e., the metamagnetic transformation exhibiting a 350 fs delay. It is associated with a change in the band structure, with the spin-minority Fe band slightly pushed below the Fermi level and getting filled by the electrons that progressively cool down. This transformation occurs on a subpicosecond timescale that is faster than what was determined by previous experiments on FeRh with a lower time resolution[45]. Most importantly, our results set a new timescale that is faster than the lattice expansion and the establishment of the macroscopic, long-range magnetic order[35–37].

The process is schematically depicted in Fig. 1b. Immediately after the action of the pulse, a significant number of electrons is excited to unoccupied states, with a non-thermal distribution of electrons $f_{n\mathbf{k}}(t)$ (red arrow in Fig. 1b) decaying toward the Fermi level on a time scale of about 200 fs. This initial stage can be considered as a precursor of the FM phase and is accompanied by the slower dynamics of the band structure $\epsilon_{n\mathbf{k}}$ with the formation of the peak of the Fe minority band. The peak crosses the Fermi level at about 350 fs delay, and stabilizes at −150 meV binding energy after 400 fs. During this step, the system undergoes a purely electronic transition through a transient phase, where the electronic band configuration evolves from $\epsilon_{n\mathbf{k}}(t)$ to an intermediate electronic FM phase $\epsilon_{n\mathbf{k}}^{e-FM}$ (blue arrow in Fig. 1b). Once the electronic distribution reaches the configuration of the FM phase (after 400 fs), the relaxation of the lattice parameter toward the equilibrium value of the FM phase then follows on a longer time scale (black arrow in Fig. 1b).

In conclusion, we determine the existence of a transient electronic phase needed to induce the AFM to FM phase transition of FeRh using pump-probe photoelectron spectroscopy at the FLASH FEL facility. The results are supported by electronic structure calculations, which explain the details of the dynamics following the laser excitation. The time-resolved photoemission experiment at a laser fluence of 5.6 mJ/cm$^2$ is well reproduced assuming the excitation of 0.25 electrons per unit cell of FeRh. At these fluences, the photon absorption cannot be described simply as promotion of electrons from filled to empty states of the calculated band structure, but the modified electron population induces a modification of the band structure as well, which is confirmed by the good agreement between theory and experiments. The laser excitation results in the transfer of electrons from the occupied $d$ orbitals below the Fermi level to the unoccupied $d$ orbitals above the Fermi level, with a partial transfer of electrons from the Rh to the Fe sites. The transient electronic phase exists up to 500 fs after the laser excitation. The emergence

of the FM phase can be followed by the appearance and position of the Fe minority band near the Fermi level with characteristic time of $\tau = 220 \pm 110$ fs. Photoexcited electrons relax across the Fermi level and establish the FM electronic band structure 400 fs after the laser excitation. We thus conclude that metamagnetism in FeRh is triggered on a subpicosecond time scale. Further exploration of the laser-induced dynamics in ultrathin and nanoscale confined FeRh[31] could lead to ultrafast devices based on magnetic order-order phase transitions at room temperature.

## Methods

**Sample and surface preparation.** The sample consists of an epitaxial 80-nm-thick FeRh(001) film grown onto a MgO(001) substrate by dc magnetron sputtering using an equiatomic target. The films were grown at 725 K and post-annealed in situ at 1070 K for 45 min in order to achieve CsCl-type chemical ordering[52]. Upon cooling down the samples in the ultra high vacuum chamber, single-layer graphene is formed on top of the FeRh surface by segregating the carbon from the film[53]. This provides oxidation protection in air and avoids the need for further capping layers of the FeRh layer to be transported to the FEL facility without degradation. The sample surface was prepared via annealing only, in order to preserve the graphene layer, and tested with XPS prior to the time-resolved experiments as described in earlier works[41,45]. Examination with low-energy electron diffraction revealed the expected reconstruction pattern of the FeRh(001) surface.

**Structural and magnetic characterization of FeRh films.** X-ray reflectivity and x-ray diffraction measurements confirm the smooth character and high-quality FeRh(001) texture of the film. The existence of the magnetic phase transition was confirmed via temperature-dependent magnetization measurements using vibrating sample magnetometry (see Supplementary Fig. 1 for sample characterization and ref. [52] for additional information on FeRh films grown under equivalent conditions).

**Experiment.** The experiments are performed at the plane grating monochromator beamline[54,55] at FLASH[56,57], using the HEXTOF end-station[58]. The pump-probe scheme is established by a near-infrared pulse of 90 fs coupled with a FEL pulse of about 130 fs (both values are the full width half maximum, FWHM), which provide an estimated system response function[58], i.e., the effective pump-probe correlation, of ~150 fs FWHM. The optical pump and x-ray probe energies were set to 1.55 eV and 123.5 eV, respectively. The energy resolution of ~150 meV is extracted from the Fermi level fit.

The sample temperature was kept at 348 K in the experimental chamber during all pump-probe photoelectron spectroscopy experiments reported here. At this temperature, the FeRh film is in the AFM phase during both the heating and cooling cycles (see Supplementary Fig. 1), which ensures the relaxation of the FeRh film back to the AFM phase between consecutive laser pulses.

Photoexcited electrons near normal emission were detected using a momentum microscope, which has an acceptance angle of $2\pi$ above the sample surface and can image the full Brillouin zone (BZ) with up to 7 Å$^{-1}$ diameter[58,59]. We used a negative extractor voltage (~40 V with respect to the sample potential). This retarding field between the sample and extractor effectively removes the slow secondary electrons originating from the x-ray photons and pump-laser-induced slow electrons. All background electrons with energies less than ~4 eV are thus repelled within the first 400 μm above the sample surface. This removal of space charge comes at the expenses of k-resolution and causes a reduction of the k-field-of-view to 1.3 Å$^{-1}$. Integrating over this k-field represents the integral of 60% of the BZ of FeRh, which was sufficient to well identify the peak associated with the FM phase. We characterize the sample surface by measuring the spectra of the system in the AFM and FM phases at fixed temperatures, and obtained line-shapes equivalent to those reported in ref. [45] (see Fig. 1d). The presence of the peak at about ~150 meV below the Fermi level is the signature of the Fe minority band characteristic of the FM phase. To fit the experimental data in Fig. 4b, we used an error function of the following form:

$$f(t) = y_0 + A\left[1 + \mathrm{erf}\left(\frac{t - t_D}{\tau}\right)\right]$$

where $y_0$ is a vertical offset, $A$ is the amplitude, $t_D$ is the temporal onset of the transition (with respect to $t_0$), and $\tau$ is the characteristic rise time.

**Theory.** We calculate from first principles the equilibrium and non–equilibrium properties of FeRh using the pw.x and yambo codes[60–63] within Density Functional Theory (DFT) and its Time Dependent (TD–DFT) extension. At equilibrium both the FM and AFM phases are computed within the local density approximation (LDA) fully including spin–orbit coupling (SOC). An energy cut-off of 65 Ry is used for the wave–functions with a $5 \times 5 \times 5$ sampling of the BZ for the self–consistent calculation. The experimental lattice parameter, 5.966 Å, is chosen for the AFM structure: an FCC unit cell containing 4 atoms (there is a factor 2

compared to the parameter of the BCC unit cell with 2 atoms used in ref. [64]). The FM ground state is then computed for the same unit cell and for a unit cell with a 1% lattice expansion. LDA gives a low negative stress using the experimental value of the lattice parameters in both the FM and the AFM phase. We use the experimental values and we verified that changes in the lattice parameters very weakly affect the electronic density of states.

We also verified that the Generalized Gradient Approximation (GGA) gives small improvements in comparison with experimental values, which however are not relevant to the present work. For this reason we used LDA which is more easily handled in the non equilibrium TD–DFT simulations with SOC. Finally we verified that the AFM structure displays a phonon instability as reported in the literature[64–66].

Subsequently, a non self–consistent calculation (NSCF) on a $8 \times 8 \times 8$ sampling of the BZ is performed. The electronic density of states (DOS) of the two structures reported in Fig. 1c is computed starting from such NSCF calculation. We then construct the XPS spectrum from the projection of the DOS on the atomic orbitals of Fe and Rh. The projected DOS are weighted using tabulated photoionization cross sections[67] for a probe of 125 eV (in practice the signal is dictated by Fe(3d) orbitals). Moreover, we use energy dependent lifetimes of the form $\gamma_{nk} = A + B\, d(\epsilon_{nk}) + C(\epsilon_F - \epsilon_{nk})^2$, where the first constant contribution $A = 60$ meV mimics the experimental resolution, the second term $B$, proportional to the electronic DOS $d(\epsilon)$, mimics the electron–phonon lifetimes, and the term which grows quadratically away from the Fermi level mimics the electron–electron lifetimes. Finally, the effect of temperature is included in the Fermi distribution used for the electronic occupations. The resulting spectrum is shown in Fig. 1c.

The TD–DFT simulations, as implemented in the yambo code[68], are then performed propagating the Kohn–Sham density matrix in the basis–set of the equilibrium wave–functions under the action of the same pump pulse used in the experiment. The NSCF DFT calculation is used as a starting point for TD–DFT. The laser pulse parameters are equivalent to the experimental conditions. In particular, the fluence is chosen considering: (i) the experimental fluence, (ii) the fact that part of the pulse is reflected by the sample, and (iii) the fact that the external field is renormalized by the induced field. The effect of both (ii) and (iii) is estimated taking into account the dielectric function of bulk FeRh. In particular, point (iii) needs to be considered since we adopt the so called transverse gauge, where the macroscopic (or $G = 0$) component of the Hartree field is subtracted from the microscopic TD–DFT equations. In the input file we set the pulse intensity of $5 \times 10^7$ kW/cm$^2$ and the FWHM of the intensity profile is 100 fs. We use a Gaussian envelope times a $\sin(\omega t)$ function with $\hbar\omega = 1.55$ eV. The code computes the fluence during the simulation corresponding to an absorbed fluence of 1.881 mJ/cm$^2$. This compares fairly well with the experimental estimate. Using $1 - R \approx 0.3138$ and a laser fluence of 5.6 mJ/cm$^2$, about $0.3138 \times 5.6$ mJ/cm$^2 \approx 1.757$ mJ/cm$^2$ is absorbed by the sample. The density matrix is calculated on a $8 \times 8 \times 8$ grid of **k** points in the BZ including all states from $-3.5$ up to 5.5 eV. The Kohn–Sham field felt by the electrons is updated at each time step during the simulation. The non-equilibrium DOS is then computed by diagonalizing the Hamiltonian evaluated from the photoexcited electron density after the action of the pump pulse. The adiabatic non-equilibrium XPS spectrum is finally computed using the same smearing used for the equilibrium spectra. The electronic occupation is obtained from the diagonal elements of the density matrix in the basis updated during the course of the simulation.

## Code availability
The data that support the findings of this study are available from the corresponding author upon reasonable request. The codes used for the ab-initio electron dynamics calculations are freely distributed under GPL license under the link https://github.com/yambo-code/yambo.

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

## Acknowledgements

This work is dedicated to Wilfried Wurth, who passed away on May 8, 2019. We acknowledge support by the scientific and technical staff of FLASH, as well as Holger Meyer and Sven Gieschen form University of Hamburg. This work was supported by the excellence cluster "The Hamburg Centre for Ultrafast Imaging—Structure, Dynamics and Control of Matter at the Atomic Scale" of the Deutsche Forschungsgemeinschaft (DFG EXC 1074) and through the SFB 925 "Lichtinduzierte Dynamik und Kontrolle korrelierter Quantensysteme" (project B2). It received funding from the EU-H2020 research and innovation program under European Union projects "MaX" Materials design at the eXascale H2020-EINFRA-2015-1 (Grant Agreement No. 824143) and "NFFA" Nanoscience Foundries and Fine Analysis-Europe H2020-INFRAIA-2014-2015 (Grant Agreement No. 654360) having benefited from the access provided by the ISM node (CNR, Italy), user-project IDs 247 and 669. We acknowledge the Deutsche Forschungsgemeinschaft (DFG, German Research Foundation)—TRR 173—268565370 (projects A02 and A05). Access to the CEITEC Nano Research Infrastructure was supported by the Ministry of Education, Youth and Sports (MEYS) of the Czech Republic under the projects CEITEC 2020 (LQ1601) and CzechNanoLab (LM2018110). We acknowledge funding from the Italian project MIUR PRIN Grant No. 20173B72NB. This work has received funding from the European Union's Horizon 2020 research and innovation program under the Marie Skłodowska-Curie and it is co-financed by the South Moravian Region under grant agreement No. 665860.

## Author contributions

F.P., V.U., J.A.A., M.G., D.S., and F.S. designed the project. F.P., D.K., M.H., S.Y.A., G.B., H.R., D.V., V.U., J.A.A., and F.S. performed the time-resolved XPS experiments and analyzed the data. M.G., D.S., and A.M. designed the theoretical approach to the problem. J.A.A. and V.U. prepared and characterized the samples. All authors discussed the results. F.P., V.U., J.A.A., D.S., M.G., and F.S. wrote the paper with contributions from all authors and critical revision from J.D., G.S., and W.W.

## Funding

## Competing interests

The authors declare no competing interests.
