## [Peer Review File · Nature Communications]

Reviewers' Comments:

Reviewer #1:

Remarks to the Author:

The authors report on a time-resolved photoemission study of the laser-induced antiferromagnetic-ferromagnetic phase transition in metallic FeRh. Using the time-resolved momentum microscope at FLASH free-electron laser the authors are able to sensitively monitor the laser-induced changes of the valence band electronic structure and of the photo-excited electrons above Fermi level. The main observation is the appearance of a minority peak close to Fermi energy within about 500 fs, which is the electronic fingerprint of establishing the ferromagnetic state. This observation is highly surprising since the emergence of long-range magnetic order and of the lattice expansion accompanying the AFM-FM phase transition is taking place on much longer time scales, as reported in numerous previous studies of this phenomenon. Supported by TD-DFT calculations, the authors correlate the measured dynamic changes of the valence band with a laser-induced charge transfer from Rh to Fe atoms; such a coherent inter-atomic electron transfer, active only during the laser pulse excitation, should dynamically change the population of minority vs majority electrons at the Rh sites (initially balanced in the AFM state) and consequently should trigger the AFM-FM phase transition.

This work reveals new and important results concerning the ultrafast metamagnetic phase transition in FeRh that should trigger a large (and renewed) interest in the ultrafast science community as to the physics of laser-driven first-order phase transitions. The manuscript is well written, the presentation of the experimental and theoretical data is clear and the arguments are reasonable. I therefore recommend the manuscript for publication after the authors do the following modifications:

1. I think the title is not accurate enough. One should add 'FeRh' in the title since only this system has been studied; also other metamagnetic phase transitions might evolve on different time-scales and might be driven by different stimuli.
2. On page 2 of the manuscript when referring to femto-magnetism research field and the relevance of FM generation across first-order phase transitions, I think it is appropriate to cite, in addition to Ref. 26, more recent reviews on femto-magnetism that discuss in more detail the ultrafast AFM-FM transitions, as for instance: P. Nemeč et al., *Nature Physics* 14, 229-241 (2018) and K. Carva et al., *Handbook of Magnetic Materials* 26, 29-463 (2017).
3. In Figure 3 one should clearly describe what is actually shown in the figure, i.e. which curves are denoting the experimental data and which ones the theory calculations.
4. Although proving enough evidence supporting the main claims of the manuscript, the authors show time-resolved PES data only for one laser fluence of 5.6 mJ/cm². Are there more PES measurements performed at different fluences? If yes, the authors should show and discuss them.
5. Using the time-resolved momentum microscope at FLASH FEL one should be able to perform spin-resolved PES measurements as well. Such measurements can provide information on the transient variation of the spin polarization and of the exchange splitting of the relevant bands in FeRh. Are such spin-, time- and momentum-resolved PES measurements already available? If not, I strongly recommend the authors to perform such measurements that will provide the complete picture of the dynamic AFM-FM transition in FeRh.
6. In Figure 2 the authors show the emergence of the minority peak in the differential PES vs pump-probe delay. At energies above Fermi level one observes the time evolution of the photo-excited Fermi-Dirac distribution function and superimposed on this the appearance of the minority peak. This peak seems to appear, or some precursor of it, already at zero delay and starts to shift in energy and grow in intensity towards longer pump-probe delay. Is this the signature of the FM state nucleation that is taking place essentially instantaneously when compared to the pump pulse duration? The authors should address and discuss this point in the manuscript.
7. In Figure 3 the authors compare the experimental PES data with the TD-DFT calculated spectra. The resulting differential curves in the bottom panel show large changes above and below Fermi level. The differential changes at binding energies of -2eV and below are due to PES intensity changes of the experimental peaks while the calculated differential changes are mainly due to a peak shift at these energies. The authors should address and discuss this discrepancy in the manuscript.
8. The labels of Figure 5 are too small and unclear. The authors should improve here.

Reviewer #2:

Remarks to the Author:

Pressacco et al. report on a combined experimental and numerical study on fs laser-induced antiferromagnetic (AFM) to ferromagnetic (FM) phase transition in FeRh. It addresses the long-standing question as to whether this laser-induced first-order phase transition is primarily of electronic origin or driven by lattice expansion, which has been around for more than 15 years (see refs. 34,35). The authors address this problem by performing time-resolved photo-electron spectroscopy, where they observe the formation of a minority band characteristic for the FM phase just below the Fermi energy, within half a ps after laser excitation. Identification of this signature is established by recording static spectra both for the FM and AFM phase, and the observed dynamics is well in line with ab-initio electron dynamics calculations.

The exciting research question combined with the novel approach and highly interesting results will receive interest both from experts in the field as well as the more generally interested reader – even though the manuscript may not be easy to follow for the latter group. I do feel that the positioning of the work is not yet optimal. Also, I noticed some possible shortcomings in supporting information, and a couple of other issues. Altogether, I consider this manuscript potentially of interest for publication in Nature Communications, after the authors have appropriately addressed the comments listed below.

1. I would strongly suggest the authors to provide more data on structural and magnetic characterization of their sample(s), e.g. in a Supplementary. They do cite external work which would indicate that high-quality FeRh can be prepared on MgO(001) (ref. 54), but none of their own data. In particular, it would be relevant to see some of their magnetometry data. Similarly, the authors do report their laser fluence (5.6 mJ/cm^2) but not the associated heating of the electron system.
2. Although the data look very convincing, it seems that everything presented is based on a single set of measurements on a single sample at a single laser fluence. Could the authors provide any information on the reproducibility of the reported effects?
3. The authors conclude that their results set a new timescale that is faster than the lattice expansion. I agree about this conclusion, but not that it is new. Already in earlier work (ref. 34,35) this was concluded. Although “discarded by subsequent works” (line 50), the disagreement between refs. 34/35 and 37-39 could potentially be explained by ref. 36, where it was suggested that local magnetization already develops at sub-ps timescale, but it may need tens of ps to build-up the full magnetization over the experimentally averaged volume. I believe that the present manuscript would increase its impact by properly discussing their findings in terms of earlier literature.
4. Figure 5 is not optimally clear to me. The figure shows photo-induced changes; I would presume taking the AFM state as a reference. However, this AFM state should have half of the Fe sites having an opposite spin density than the other. Thereby, I do not understand the periodicity of the final photoinduced changes, where such a sub-lattice dependence is not seen at all. Also, I do not understand the scale of the legend. What does $1.0e-05$ mean? I can hardly believe that those are relative (dimensionless) changes, because of the large laser fluence used.
5. Apart from the formation of the minority feature just below the Fermi level, there seems to be an increase in spectral weight around 1.2 eV below E_F , and a minor decrease around 0.5 eV below E_F (Fig. 2b, + 1 ps), which would both match with features in the static X-ray spectra in the two phases (Fig. 1d). I may have overlooked it (or maybe the authors tried to address in lines 97 – 108), but I believe the authors do not convincingly address this correspondence, which looked very noteworthy to me. Could the authors comment?
6. Minor issue (maybe related to the previous point): I do not understand the “must” in line 111.
7. Minor issue: the authors refer a couple of times to two-photon absorption processes. It does not get clear whether this is of significant relevance for their main observation (the electronically driven phase transition). Also, the relative importance of the two-photon process should be depending on laser-fluence. Has this been verified?

Reviewer #3:

Remarks to the Author:

In the present work Pressacco et al. have performed a joint theory experiments work to demonstrate a AFM to FM phase transition in FeRh initiated by recently proposed OIST mechanism. The work is interesting and quite neatly demonstrate the intricacies of the phase transition. However, before recommending it for publication following issues require clarification and changes to the manuscript:

The theory clearly demonstrate that the phase transition is triggered by the OIST process. However, the authors shy away from boldly saying it in the manuscript. It is somehow toned down. Is there a reason for that? Are the authors not sure of the data or of the accuracy of the method? In either case it should be clearly said in the manuscript.

The fluence is chosen to mimic experiments. Would be good to see a direct comparison between theoretically used fluence, experimental incident and absorbed fluence. These three numbers would give an estimate of the scaling required theoretically to mimic experimental data. Without this I consider the manuscript incomplete.

The theoretical pump laser parameters should be clearly stated in the main part of the manuscript. I would like to see the FWHM, Fluence, central frequency and intensity of the pulse. This should be clearly compared to the corresponding experimental parameters. For now this part is glossed over and rather opaquely stated in the manuscript. One does not expect theory and experimental parameters to be exactly the same, given several missing contributions theoretically.

Point-by-point response to the referee reports

Reviewer #1

The authors report on a time-resolved photoemission study of the laser-induced antiferromagnetic - ferromagnetic phase transition in metallic FeRh. Using the time-resolved momentum microscope at FLASH free-electron laser the authors are able to sensitively monitor the laser-induced changes of the valence band electronic structure and of the photo-excited electrons above Fermi level. The main observation is the appearance of a minority peak close to Fermi energy within about 500 fs, which is the electronic fingerprint of establishing the ferromagnetic state. This observation is highly surprising since the emergence of long-range magnetic order and of the lattice expansion accompanying the AFM-FM phase transition is taking place on much longer time scales, as reported in numerous previous studies of this phenomenon. Supported by TD-DFT calculations, the authors correlate the measured dynamic changes of the valence band with a laser-induced charge transfer from Rh to Fe atoms; such a coherent inter-atomic electron transfer, active only during the laser pulse excitation, should dynamically change the population of minority vs majority electrons at the Rh sites (initially balanced in the AFM state) and consequently should trigger the AFM-FM phase transition.

This work reveals new and important results concerning the ultrafast metamagnetic phase transition in FeRh that should trigger a large (and renewed) interest in the ultrafast science community as to the physics of laser-driven first-order phase transitions. The manuscript is well written, the presentation of the experimental and theoretical data is clear and the arguments are reasonable. I therefore recommend the manuscript for publication after the authors do the following modifications:

We thank the referee for the overall positive evaluation of the manuscript and for highlighting the relevance and novelty of our work, as well as for his/her constructive criticism and valuable suggestions. We have addressed these comments to improve the manuscript.

1. I think the title is not accurate enough. One should add ‘FeRh’ in the title since only this system has been studied; also other metamagnetic phase transitions might evolve on different time-scales and might be driven by different stimuli.

We agree with the referee and have thus modified the title of the manuscript, which now reads: “Subpicosecond metamagnetic phase transition in FeRh driven by non-equilibrium electron dynamics”.

2. On page 2 of the manuscript when referring to femto-magnetism research field and the relevance of FM generation across first-order phase transitions, I think it is appropriate to cite, in addition to Ref. 26, more recent reviews on femto-magnetism that discuss in more detail the ultrafast AFM-FM transitions, as for instance: P. Nemeč et al., *Nature Physics* 14, 229-241 (2018) and K. Carva et al., *Handbook of Magnetic Materials* 26, 29-463 (2017).

We have followed the referee’s suggestion to cite more up-to-date review articles concerning femtomagnetism. We have included them in the revised manuscript, together with Ref. 26 (see page 2).

3. In Figure 3 one should clearly describe what is actually shown in the figure, i.e. which curves are denoting the experimental data and which ones the theory calculations.

A legend was added to Figure 3 and the figure caption was modified as follows:

“Experimental spectra (dots) at equilibrium ($t = -0.5$ ps, blue) and at the maximum of the pump pulse ($t = 0$ ps, red) are compared in linear (top panel) and logarithmic scales (inset) to the TD-DFT calculations (solid lines) of the electronic structure of FeRh in the AFM phase (blue) and at laser excitation (red). The bottom panel exhibits the difference between the corresponding experimental and theoretical spectra in the top panel.”

4. Although proving enough evidence supporting the main claims of the manuscript, the authors show time-resolved PES data only for one laser fluence of 5.6 mJ/cm^2 . Are there more PES measurements performed at different fluences? If yes, the authors should show and discuss them.

We thank the referee for bringing up this point. We have actually performed fluence-dependent PES measurements, and have now included the corresponding pump-probe photoemission spectra for a set of different fluence values in the Supplementary Information (see Supplementary Figure 2). Although not all datasets have the same high level of signal-to-noise ratio, we can identify the existence of a threshold fluence to induce the AFM-FM phase transition in our time-resolved photoemission data (in line with previous literature based on time-resolved MOKE or X-ray diffraction).

While the PES datasets for different fluences do not modify the main message of our work, we have included a brief discussion on the fluence dependence of the process in the revised manuscript (see pages 5-6). In addition, we report time-resolved PES measurements in a longer time-delay window (up to 24 ps after the pump pulse) to confirm the establishment of the FM phase at longer time scales.

5. Using the time-resolved momentum microscope at FLASH FEL one should be able to perform spin-resolved PES measurements as well. Such measurements can provide information on the transient variation of the spin polarization and of the exchange splitting of the relevant bands in FeRh. Are such spin-, time- and momentum-resolved PES measurements already available? If not, I strongly recommend the authors to perform such measurements that will provide the complete picture of the dynamic AFM-FM transition in FeRh.

The referee brings a very interesting point here, in that obtaining the spin-resolved band structure of FeRh during laser excitation would offer unprecedented insight into the identification of the relevant electronic processes involved in the laser-induced metamagnetic phase transition. As the referee suggests, such experiments are in principle possible thanks to the relatively recent advances in momentum microscopy [see, for instance, Tusche et al., *Ultramicroscopy* 159, 520-529 (2015)].

We have in fact attempted to perform such experiments using momentum-microscopy at FLASH FEL, but the joint time- and spin-resolved PES signal and corresponding count rate showed to be about two-orders of magnitude lower than the spin-integrated one within our experimental conditions. For this reason, we could not accommodate spin-resolved PES experiments at FLASH FEL with sufficiently good signal-to-noise ratio during the experiments we have performed so far. One limiting factor in every time-resolved MM is the laser induced space-charge. For future experiments we envisage to use the next development of the electronic lenses of the momentum microscope which will allow to effectively suppress the space charge, and hence widen the range of usable FEL intensity [see, for instance, Schönhense et al., *Rev. Sci. Instrum.* 92, 053703 (2021)].

All of the above in consideration, we thank the referee for the valuable advice and we envision further experimental efforts on FeRh towards time- and spin-resolved band mapping using FEL or high-harmonic generation-based sources [see, for instance, Bühlmann et al., *Rev. Sci. Instrum.* 91, 063001 (2020)], which could assist in providing the longer acquisition time conditions that are necessary for this experiment.

6. In Figure 2 the authors show the emergence of the minority peak in the differential PES vs pump-probe delay. At energies above Fermi level one observes the time evolution of the photo-excited Fermi-Dirac distribution function and superimposed on this the appearance of the minority peak. This peak seems to appear, or some precursor of it, already at zero delay and starts to shift in energy and grow in intensity towards longer pump-probe delay. Is this the signature of the FM state nucleation that is taking place essentially instantaneously when compared to the pump pulse duration? The authors should address and discuss this point in the manuscript.

The referee brings here an interesting question on whether we can argue about the existence of an “instantaneous” laser-induced AFM-to-FM electronic phase transition. Indeed, the $t = 0$ ps energy-dependent PES data set in Figure 2b exhibits the existence of a transient

peak-like DOS feature just above the Fermi level ($E - E_F \sim 0.2$ eV). However, we attribute this initial localized increase of the DOS to the electronic filling of the empty Fe bands corresponding to the AFM electronic band structure, as well as to the band broadening associated with the presence of the optical pump. Despite this filling happening immediately after the laser pump, our results show that the band structure modification that can be directly linked to the metamagnetic phase transition only happens about 300-to-400 fs later, which is manifested by the crossing of the DOS peak from above to below the Fermi level, a process finally giving rise to the FM electronic band structure which is equivalent to the high-temperature static PES spectrum (see Figure 1d).

The referee is right in that one can assign the role of a phase-transition precursor to this immediate pump-induced increase in DOS just above the Fermi level. However, we cannot affirm that the FM phase nucleation happens at the same time scale as the optical pump. During this intermediate temporal region, the system cannot be linked to any of the equilibrium AFM or FM electronic states, such that we believe it would be correct to talk about the appearance of a precursor at $t = 0$, but not of the FM phase nucleation, strictly speaking. The occurrence of the metamagnetic phase transition is manifested in the electronic band modification, which can only explain a build-up of the DOS peak above the Fermi level, persistent on a timescale well above one picosecond. In the absence of this band modification, de-excited electrons would simply accommodate back in the deeper bands from which they were promoted at $t = 0$.

We have clarified this aspect in the revised version of the manuscript (see pages 6, 8).

7. In Figure 3 the authors compare the experimental PES data with the TD-DFT calculated spectra. The resulting differential curves in the bottom panel show large changes above and below Fermi level. The differential changes at binding energies of -2eV and below are due to PES intensity changes of the experimental peaks while the calculated differential changes are mainly due to a peak shift at these energies. The authors should address and discuss this discrepancy in the manuscript.

Both in the experimental data and in the numerical simulation the broad peak at ≈ -2 eV is affected by the out-of-equilibrium density right after the pump, i.e. at $t = 0$. Since we rule out two-photon absorption processes (their intensity is nearly two orders of magnitude smaller), it is not possible to explain such signal in terms of a change in the electronic occupations $f_{n\mathbf{k}}$ directly induced by the laser pulse. Moreover, scattering processes can also be ruled out since the signal has no time delay compared to the pulse (see Figure 4). A change in the band structure $\epsilon_{n\mathbf{k}}$ is the only possible explanation. This is what the numerical simulations confirm. Theory and experiments agree in a reduction of the electron yield in the energy region $E < -1.55$ eV as discussed in the main text.

As the reviewer points out, there is a difference between theory and experimental data in the energy region $-1.55 < E < 0$: in the numerical simulation the changes in the band structure appears as an overall shift of the broad peak at ≈ -2 eV, while in the experimental data the effect shows as a reduction of the broad peak intensity. Our explanation of this difference is the following. Experimentally the change in the occupations $f_{n\mathbf{k}}$ compensates the change in the band structure $\epsilon_{n\mathbf{k}}$ in the energy region $-1.55 < E < 0$, giving an ≈ 0 signal. Such compensation is not exact in the numerical simulations, giving a weak non-zero signal. This is likely due to the fact that scattering processes are neglected (see also the differences above the Fermi level). A less peaked distribution of the occupations could give better agreement and turn the overall effect into a peak intensity reduction also starting from the numerical simulations. We underline that such cancellation is needed to explain the ≈ 0 experimental signal in the range of binding energies from which electrons are excited from by the laser pulse.

We have rephrased the discussion in the main text to better clarify this point (page 5).

8. The labels of Figure 5 are too small and unclear. The authors should improve here.

We have increased the label size for better clarity.

Reviewer #2

Pressacco et al. report on a combined experimental and numerical study on fs laser-induced antiferromagnetic (AFM) to ferromagnetic (FM) phase transition in FeRh. It addresses the long-standing question as to whether this laser-induced first-order phase transition is primarily of electronic origin or driven by lattice expansion, which has been around for more than 15 years (see refs. 34,35). The authors address this problem by performing time-resolved photo-electron spectroscopy, where they observe the formation of a minority band characteristic for the FM phase just below the Fermi energy, within half a ps after laser excitation. Identification of this signature is established by recording static spectra both for the FM and AFM phase, and the observed dynamics is well in line with ab-initio electron dynamics calculations.

The exciting research question combined with the novel approach and highly interesting results will receive interest both from experts in the field as well as the more generally interested reader – even though the manuscript may not be easy to follow for the latter group. I do feel that the positioning of the work is not yet optimal. Also, I noticed some possible short comings in supporting information, and a couple of other issues. Altogether, I consider this manuscript potentially of interest for publication in Nature Communications, after the authors have appropriately addressed the comments listed below.

We thank the referee for pointing that we are addressing an interesting scientific question here and for saying that we bring highly interesting results in our manuscript. We have now tried to amend the shortcomings mentioned by the referee in the report. We are grateful for his/her suggestions and questions, and we believe they have significantly helped to improve the manuscript.

1. I would strongly suggest the authors to provide more data on structural and magnetic characterization of their sample(s), e.g. in a Supplementary. They do cite external work which would indicate that high-quality FeRh can be prepared on MgO(001) (ref. 54), but none of their own data. In particular, it would be relevant to see some of their magnetometry data.

Following the recommendation of the referee, we have now added structural (XRR, XRD) and magnetic characterization (VSM) of the FeRh film employed in the photoemission experiments at FLASH (see Supplementary Figure S1). The sample characterization details have also been added to the “Methods” section (page 10). We would like to mention that the citation concerning the growth and characterization of high-quality FeRh films (ref. 54 in the original manuscript, ref. 55 in the revised version) consists of our own previous work, where more detailed structural and magnetic properties of FeRh films grown in equivalent conditions are reported.

Similarly, the authors do report their laser fluence (5.6 mJ/cm^2) but not the associated heating of the electron system.

We have estimated the temperature of the electronic system by fitting the pump-probe photoelectron spectra to a Fermi-like distribution function (see our earlier work, ref. 48). As an exemplary value, we obtain an electronic temperature amounting to $T_e \sim 1250 \text{ K}$ shortly after the laser pulse ($t = 0.3 \text{ ps}$) for a fluence of 4.5 mJ/cm^2 .

2. Although the data look very convincing, it seems that everything presented is based on a single set of measurements on a single sample at a single laser fluence. Could the authors provide any information on the reproducibility of the reported effects?

The referee is right here and we have thus added additional pump-probe photoemission results in the Supplementary Information file, including fluence-dependent data which prove the consistency of our results.

Specifically, Figure S2(a) shows photoelectron spectra obtained before ($t < 0$) and after ($t = 2 \text{ ps}$) the laser pulse for four different values of the laser fluence. Above a threshold

fluence we appreciate the presence of the minority band peak formation 2 ps after the laser pulse, thus indicating a persistent modification of the band structure and the metamagnetic phase transition. In Figure S2(b) we show the differential photoemission intensity map plotted in the same fashion as Figure 4a, where the presence of the minority phase is followed on a longer time delay up to 24 ps after the laser pulse (laser fluence, 4.5 mJ/cm²). The panels S2(c) and S2(d) present differential photoelectron spectra at selected delays as in Figure 2b for 1.1 and 4.5 mJ/cm² fluences values, respectively.

We believe that these supplementary data sets add to the consistency of our results and offer further insight into the specifics of the electron dynamics across the laser-induced phase transition of FeRh. We have modified the manuscript accordingly to briefly discuss the aspects related to experiments and different fluences (see pages 5-6).

3. The authors conclude that their results set a new timescale that is faster than the lattice expansion. I agree about this conclusion, but not that it is new. Already in earlier work (ref. 34,35) this was concluded. Although “discarded by subsequent works” (line 50), the disagreement between refs. 34/35 and 37-39 could potentially be explained by ref. 36, where it was suggested that local magnetization already develops at sub-ps timescale, but it may need tens of ps to build-up the full magnetization over the experimentally averaged volume. I believe that the present manuscript would increase its impact by properly discussing their findings in terms of earlier literature.

We thank the referee for this valuable comment concerning the earlier literature on the laser-induced phase transition in FeRh. We agree with the fact that ref. 36 (ref. 38 in the revised manuscript version) already introduced the concept of FM domain nucleation happening at subpicosecond time scales. This work argued that the delayed perception of the transition when using MOKE or XMCD probes (sensitive to magnetization alignment) is caused by the additional time interval the system needs to exhibit field-induced FM domain growth and coalescence leading to a nonzero net magnetization. Our results are fully compatible with this scenario and confirm the subpicosecond appearance of the FM electronic order parameter.

We have now rephrased this introductory paragraph in order to discuss the existing literature and the corresponding context of our work in due form (see pages 2-3).

4. Figure 5 is not optimally clear to me. The figure shows photo-induced changes; I would presume taking the AFM state as a reference. However, this AFM state should have half of the Fe sites having an opposite spin density than the other. Thereby, I do not understand the periodicity of the final photoinduced changed, where such a sub-lattice dependence is not seen at all. Also, I do not understand the scale of the legend. What does 1.0e-05 mean? I can hardly believe that those are relative (dimensionless) changes, because of the large laser fluence used.

The charge and spin density units are in atomic units as written in the caption, i.e. [e/a_0^3] and [μ_B/a_0^3], where $a_0 = 5.29 \times 10^{-11}$ m is the Bohr radius. 0.25 electron per unit cell corresponds to a density of $7 \times 10^{-4} e/a_0^3$. The fact that such electronic density is excited from valence to conduction states does not imply that there is locally the same change in real space, since valence and conduction orbitals have a strong spatial overlap. Indeed, the computed 3D density variation (from which the 2D cut is extracted) has a maximum of $\approx 10^{-3} e/a_0^3$ and the average density variation is much smaller.

The periodicity of Fe atoms in the AFM structure is seen in Fig. 5(c) where a 2D cut in the plane of the Rh atoms is chosen. Blue (negative) spots are for the Fe atoms in the “above plane”, while red (positive) spots are for the (non visible) Fe atoms in the “below plane”. The effect of the laser excitation is that “the local Fe spin density is reduced”. So for the atoms that have “positive” spin magnetization in the ground state, the laser-induced spin magnetization “change is negative” (and vice versa).

5. Apart from the formation of the minority feature just below the Fermi level, there seems to be an increase in spectral weight around 1.2 eV below E_F , and a minor decrease around 0.5 eV below E_F (Fig. 2b, + 1 ps), which would both match with features in the static X-ray spectra in the two phases (Fig. 1d). I may have overlooked it (or maybe the authors

tried to address in lines 97 – 108), but I believe the authors do not convincingly address this correspondence, which looked very noteworthy to me. Could the authors comment?

We thank the referee for highlighting the correspondence between the pump-probe spectra for $t > 0.5$ ps (Figure 2b) and the static spectrum at 420 K (Figure 1d). Our results reveal in fact a remarkable agreement in a wider energy region down to ~ 1.5 eV below the Fermi level, beyond the mere formation of the minority peak near E_F . In the previous version of the manuscript, we had focused our discussion mainly to the formation of the minority band feature. However, we agree that it is worth discussing these more subtle aspects, as it aggregates further evidence of the subpicosecond establishment of the FM phase upon laser excitation.

We have now added a brief part in the discussion of Figure 2b (page 5) in the revised version of the manuscript in order to underline this feature.

6. Minor issue (maybe related to the previous point): I do not understand the “must” in line 111.

We have rephrased this sentence to better clarify the use of the word “must” (page 5).

7. Minor issue: the authors refer a couple of times to two-photon absorption processes. It does not get clear whether this is of significant relevance for their main observation (the electronically driven phase transition). Also, the relative importance of the two-photon process should be depending on laser-fluence. Has this been verified?

The discussion on the two-photon signal is relevant for the comparison between numerical simulations and experimental data and the analysis of the signal in the region below -1.55 eV. Indeed establishing that the signal due to two-photon processes is very low (and in agreement between theory and experiment) we can rule it out as a possible explanation for the signal at about -2 eV. We believe instead it is not particularly relevant for the phase transition, since the associated induced changes in the electronic density are very small.

The relative importance of one photon- and two-photon processes has been verified in the numerical simulations. As expected the photoexcited electron intensity due to two-photon processes increases quadratically with the laser pulse intensity.

Reviewer #3

In the present work Pressacco et al. have performed a joint theory experiments work to demonstrate a AFM to FM phase transition in FeRh initiated by recently proposed OIST mechanism. The work is interesting and quite neatly demonstrate the intricacies of the phase transition. However, before recommending it for publication following issues require clarification and changes to the manuscript:

We thank the referee for highlighting the novelty and high interest of our work. We have now attempted to clarify and amend the aspects raised in the report in order to improve the manuscript.

The theory clearly demonstrate that the phase transition is triggered by the OIST process. However, the authors shy away from boldly saying it in the manuscript. It is somehow toned down. Is there a reason for that? Are the authors not sure of the data or of the accuracy of the method? In either case it should be clearly said in the manuscript.

We agree with the referee. Our analysis clearly shows that the phase transition is initiated by the change of Fe-Rh hybridization and the Fe-Fe “optical induced spin transfer” (OIST) mechanism. Our conclusion is that this process is necessary, however not sufficient alone for realising the AFM-FM phase transition. Indeed, the immediate result of the optical excitation is the reduction of the Fe magnetic moments, but the system is still in the AFM phase.

The text in the manuscript clarifying our conclusion now reads (page 8): “Our simulations therefore suggest that the change of Fe-Rh hybridization ($Rh \rightarrow Fe$ process) and the intersite optical induced spin transfer (OIST) ($Fe(\uparrow) \leftrightarrow Fe(\downarrow)$ process) play a crucial role in the photo-induced transition. The OIST process has been recently proposed, on the basis of TD-DFT simulations, as a key mechanism also in other multicomponent magnetic materials⁵⁰⁻⁵². Here, it causes to weaken the AFM ordering but is not sufficient to trigger the magnetic transition alone (just after the photoexcitation, the system is still in the AFM phase).”

The fluence is chosen to mimic experiments. Would be good to see a direct comparison between theoretically used fluence, experimental incident and absorbed fluence. These three numbers would give an estimate of the scaling required theoretically to mimic experimental data. Without this I consider the manuscript incomplete.

The theoretical pump laser parameters should be clearly stated in the main part of the manuscript. I would like to see the FWHM, Fluence, central frequency and intensity of the pulse. This should be clearly compared to the corresponding experimental parameters. For now this part is glossed over and rather opaquely stated in the manuscript. One does not expect theory and experimental parameters to be exactly the same, given several missing contributions theoretically.

We have followed the recommendation of the referee to include this information in a comprehensive way in the manuscript. Below, we report the data for the pulse used in the simulations:

We use a pulse intensity of 5×10^7 kW/cm² and the FWHM of the intensity profile is 100 fs. We use a Gaussian envelop times a $\sin(\omega t)$ function with $\omega = 1.55$ eV. The code computes the fluence during the simulation giving 1.881 mJ/cm². This compares fairly well with the experimental estimate. Using $1 - R \approx 0.3138$ (obtained from the dielectric function of FeRh $\epsilon(\omega)$ at 1.55 eV), of 5.6 mJ/cm², about 0.3138×5.6 mJ/cm² = 1.75728 mJ/cm² is absorbed in the sample.

These parameters have been added into the “Methods” section of the manuscript (page 11).

Reviewers' Comments:

Reviewer #1:

Remarks to the Author:

In the revised version of the manuscript the authors have satisfactorily addressed my criticism and comments. I therefore endorse the publication of the revised manuscript in Nature Communications.

Reviewer #2:

Remarks to the Author:

I thank the authors for very carefully addressing my comments, and adapting their manuscript accordingly. I highly appreciate the additional data and clarifications. I have no further objections, and recommend publication as is.

Reviewer #3:

Remarks to the Author:

The manuscript is now ready for publication. All my comments have been taken care of.